# Enhancing Dissolution and Oral Bioavailability of Ursodeoxycholic Acid with a Spray-Dried pH-Modified Extended Release Formulation

**DOI:** 10.3390/pharmaceutics14051037

**Published:** 2022-05-11

**Authors:** Jaehyeok Lee, Chul Haeng Lee, Jong-Geon Lee, So Yeon Jeon, Min-Koo Choi, Im-Sook Song

**Affiliations:** 1BK21 FOUR Community-Based Intelligent Novel Drug Discovery Education Unit, Vessel-Organ Interaction Research Center (VOICE), Research Institute of Pharmaceutical Sciences, College of Pharmacy, Kyungpook National University, Daegu 41566, Korea; here0723@gmail.com (J.L.); jkl7604@naver.com (J.-G.L.); 2College of Pharmacy, Dankook University, Cheonan-si 31116, Korea; hang1130@naver.com (C.H.L.); ojsw97@naver.com (S.Y.J.); minkoochoi@dankook.ac.kr (M.-K.C.)

**Keywords:** ursodeoxycholate (UDCA), oral bioavailability, pH-modified extended release formulation, spray-drying method

## Abstract

Ursodeoxycholate (UDCA) has low oral bioavailability and pH-dependent solubility and permeability. Thus, we developed a pH-modified extended-release formulation of UDCA using Na_2_CO_3_ as the alkalizing agent and hydroxypropyl methylcellulose (HPMC) as the release-modifying agent. The optimized pH-modified controlled-release UDCA formulation, with the UDCA:HPMC:Na_2_CO_3_ ratio of 200:600:150 (*w*/*w*/*w*), was prepared using a spray-drying method. Then, the formulation’s solubility, dissolution, and pharmacokinetic properties were characterized. In a pH-modified extended-release formulation of UDCA, the solubility of UDCA was increased to 8 mg/mL with a sustained dissolution for 12 h. Additionally, the spray-dried formulation exhibited amorphous states without molecular interaction among UDCA, Na_2_CO_3_, and HPMC. Moreover, the plasma UDCA concentration of the formulation maintained a higher UDCA concentration for up to 48 h than that of UDCA itself or the non-extended-release UDCA formulation. Consequently, the formulation significantly increased the AUC compared to UDCA or the non-extended-release UDCA formulation in rats. In conclusion, we have improved UDCA’s solubility and dissolution profile by preparing a pH-modified extended-release formulation with the UDCA:HPMC:Na_2_CO_3_ ratio of 200:600:150 (*w*/*w*/*w*), which effectively increased the oral bioavailability of UDCA by 251% in rats.

## 1. Introduction

Ursodeoxycholate (UDCA) (Figure 1), an endogenous hydrophilic bile salt abundant in bear bile, has been used as a traditional medicine to treat jaundice. In 1989, the therapeutic efficacy of UDCA was demonstrated for the first time in clinical trials in patients with primary biliary cholangitis. As a result, UDCA has been marketed as a therapeutic for cholestasis and a preventive drug for liver diseases [1,2,3]. For example, Ursa^®^ (Daewoong Pharmaceutical Co., Ltd., Seoul, Korea), a single tablet with 100–300 mg of UDCA, has been marketed in Korea since 1961 to cure liver diseases, including cholestasis. Presently, UDCA is the most widely prescribed drug for the treatment of cholestasis. In addition, it has been the only drug approved by the US Food and Drug Administration for primary biliary cirrhosis since 1997 [3].

The application of UDCA extends to the treatment of non-cholestatic liver diseases, owing to its multiple modes of action, such as reducing the serum levels of toxic hydrophobic bile salts [4], stimulating the hepatobiliary excretion of xenobiotics via phase II and III detoxification processes [3,5], having antioxidant activity against oxidative stress [6,7], and exerting anti-apoptotic effects on signaling pathways, such as protein kinase C and mitogen-activated protein kinases (MAPKs) [8,9,10].

Despite its extended usage, UDCA has unfavorable physicochemical properties. For example, UDCA is practically insoluble in water with the aqueous solubility of 160 μg/mL [11]; it also has low absorption profile and bioavailability [12]. The absorption of UDCA mainly occurs at the jejunum and ileum [12]. However, the intestinal absorption of UDCA was incomplete, reaching approximately 47.7% ± 9.0% for a 500 mg oral dose, and it even decreased with an increased dose, at 19.6% ± 9.1% for a 1000 mg oral dose [13]. In addition, the plasma concentration profile of UDCA varied in its solubilization in the gastrointestinal tract in seven human volunteers [14]. The results suggest the importance of higher pH maintenance at the absorption site of UDCA.

Meanwhile, the solubility of UDCA is noticeably increased under high pH conditions [12], as it reduced crystallinity and micronized particles [11]. In addition, a 2-hydroxypropyl-β-cyclodextrin-inclusion complex of UDCA significantly increases UDCA’s dissolution profile and area under the plasma concentration curve (AUC) in human volunteers [15]. Moreover, other UDCA formulations have been developed. For example, by preparing the UDCA submicron emulsion loaded with a UDCA–phospholipid complex, the bioavailability (BA) of UDCA was increased by 374% [16]. In addition, using a UDCA-phospholipid complex increases the oral BA by 2.4-fold in rats compared with a UDCA suspension [17]. These results suggest that the increased solubility and logP of a UDCA-phospholipid complex helps increase the oral BA of UDCA.

Subsequently, Scalia et al. found that although the cross-linked sodium carboxymethylcellulose in UDCA-XL tablets increased the dissolution profile of UDCA by 2-fold, there was no significant difference in oral BA between the 300 mg UDCA-XL tablets and the 300 mg UDCA reference capsules in six human subjects. However, applying an enteric coating on UDCA-XL tablets increased their oral BA by 170% [18]. In addition, significantly increased plasma concentrations and delayed absorption time were obtained from UDCA in extended-release capsules compared to an immediate-release formulation of UDCA [19]. Moreover, according to Simoni et al. [20], enteric coated sinking UDCA tablets have a significantly increased oral BA in 12 healthy subjects compared with conventional UDCA gelatin capsules.

Taken together, these results suggest that the release of UDCA at the absorption site is important for enhancing its oral BA, and the sustained release of an alkalizing agent appears necessary to enhance the solubility of UDCA for a long period. To prove our hypothesis, we investigated pH dependency in the intestinal permeability of UDCA in Caco-2 cells as well as its solubility in the present study. To correlate in vitro pH-dependent solubility and permeability with in vivo oral BA, we investigated the effect of pH modulators on the in vivo pharmacokinetics of UDCA in rats. In addition, we prepared a pH-modified controlled-release UDCA formulation with Na_2_CO_3_ as the alkalizing agent and hydroxypropyl methylcellulose (HPMC) as the controlled-release modulator [21,22], using a spray-drying method. The formulation was optimized with its solubility and dissolution profile. Finally, we aimed to characterize the pharmacokinetics and oral BA of the optimized formulation in rats.

## 2. Materials and Methods

### 2.1. Materials

Ursodeoxycholate (UDCA) (Daewoong Pharmaceutical Co., Ltd., Seoul, Korea) and hydroxypropyl methylcellulose (HPMC) (Acros Organics, Geel, Belgium) were obtained. In addition, Naringenin (internal standard; IS), Hank’s balanced salt solution (HBSS, adjusted to pH 5.4 using 2-(N-morpholino) ethanesulfonic acid or to pH 7.4 using Tris-base), sodium citrate, NaOH, and Na_2_CO_3_ (Sigma–Aldrich Chemical Co., St. Louis, MO, USA) were purchased. Moreover, Dulbecco’s Modified Eagle Medium, fetal bovine serum, non-essential amino acids, collagen-coated 12-transwell, and penicillin–streptomycin (Corning Inc., Corning, NY, USA) were procured. All other chemicals and solvents were reagent or analytical grade.

### 2.2. pH-Dependent Solubility of UDCA

Ten mg of UDCA was mixed in 2 mL buffers of varying pH (i.e., 0.1 M HCl buffer at pH 1.2, 0.1 M acetate buffer at pH 4.0, 0.1 M phosphate buffer at pH 5.5 and at pH 7.5), and the mixture was incubated for 6 h at 25 °C. Then, the mixture was filtered through a 0.45 μm membrane filter and diluted 100-fold using 50% methanol. The concentration of UDCA in the diluted filtrates was analyzed using a liquid chromatography-tandem mass spectrometry (LC-MS/MS) system.

### 2.3. pH-Dependent Permeability of UDCA

Caco-2 cells (passage no 41–43; purchased from the American Type Culture Collection, Rockville, MD, USA) were grown in tissue culture flasks containing Dulbecco’s modified eagle medium supplemented with 20% fetal bovine serum, 1% non-essential amino acids, and 1% penicillin–streptomycin. Caco-2 cells were seeded on collagen-coated 12-transwell membranes at a density of 5 × 10^5^ cells/mL and maintained at 37 °C in a humidified atmosphere with 5% CO_2_/95% air for 21 days. The culture medium was replaced every other day. On the day of the experiment, the growth medium was discarded, and the attached cells were washed with pre-warmed HBSS (pH 7.4 or pH 5.4) and preincubated with HBSS for 20 min at 37 °C, and the permeability assay was conducted as previously described [23,24,25]. Briefly, to measure the apical to basal permeability (P_app,AB_) of UDCA at pH 5.4 or pH 7.4, 0.5 mL of HBSS (pH 7.4 or pH 5.4) containing 20 µM of UDCA was added to the apical side (inside of the insert), and 1.5 mL of fresh HBSS (pH 7.4 or pH 5.4) was added to the basal side of the insert. The insert was transferred to a well containing 1.5 mL of fresh HBSS (pH 7.4 or pH 5.4) every 15 min for 1 h. Aliquots (0.1 mL) in the basal side were transferred to clean tubes and stored at −80 °C until further analysis.

To measure the basal to apical permeability (P_app,BA_) of UDCA at pH 5.4 or pH 7.4, 1.5 mL of HBSS (pH 7.4 or pH 5.4) containing 20 µM of UDCA was added to the basal side (bottom of the insert), and 0.5 mL of fresh HBSS (pH 7.4 or pH 5.4) was added to the apical side of the insert. Aliquots (0.3 mL) from the apical side were collected and compensated with an equal volume of pre-warmed fresh HBSS (pH 7.4 or pH 5.4) every 15 min for 1 h. Samples were stored at −80 °C until further analysis. The concentrations of UDCA in the samples were analyzed using an LC-MS/MS system. The transepithelial electrical resistance (TEER) values in the Caco-2 cell system were measured before and after the experiments using an epithelial volt/ohm meter (World Precision Instruments; Sarasota, FL, USA) to monitor the integrity of Caco-2 cell monolayers. The permeability of 2 µM propranolol (a marker for high permeability and transcellular pathway) and 50 µM atenolol (a marker for low permeability and paracellular pathway) were measured, as previously described [26].

### 2.4. Optimization of UDCA Formulation

#### 2.4.1. Optimization of HPMC

Dissolution studies were conducted in 900 mL of distilled water for 5 h in a D-63150 dissolution test apparatus (Erweka, Heusenstamm, Germany) at 37 °C and 50 rpm using a paddle method (a type 2 USP dissolution method). Briefly, spray-dried formulation of UDCA with different amounts of HPMC (100, 200, and 600 mg) and 200 mg of UDCA and 10 mg of NaOH were packaged into a hard gelatin capsule (size No. 0) and placed inside a sinker. Then, 1 mL aliquots were collected from the medium at 0, 20, 40 min, 1, 2, 3, 4, and 5 h and filtered using a 0.45 μm membrane filter. Meanwhile, the medium was replenished with an equal volume of water after each sampling. Lastly, the UDCA concentrations in the filtrates were analyzed using an LC-MS/MS system.

#### 2.4.2. Optimization of Na_2_CO_3_

Ten mg of UDCA was mixed in 2 mL of distilled water with increasing concentrations of NaOH or Na_2_CO_3_ at the final concentrations of 0, 0.01, 0.05, 0.1, 0.5, 1, and 5 mg/mL. The mixture was incubated at 25 °C for 6 h. The mixture was filtered through a 0.45 μm membrane filter and diluted 1000-fold in 50% methanol. The UDCA concentrations in the diluted filtrates were analyzed using an LC-MS/MS system.

We also measured the solubility of the spray-dried UDCA formulations. Powders of the spray-dried formulations containing 10 mg of UDCA were mixed in 2 mL of distilled water and incubated at 25 °C for 6 h. The mixture was filtered through a 0.45 μm membrane filter and was diluted 1000-fold in 50% methanol. The UDCA concentrations in the diluted filtrates were analyzed using an LC-MS/MS system. Dissolution studies were also conducted in 900 mL of distilled water at 37 °C for 12 h using the previously described method. In summary, spray-dried formulation powders that contained 20 mg of UDCA were packaged into a hard gelatin capsule (size No. 0) and placed inside a sinker. A 1 mL aliquot of a medium was collected at 0, 1, 2, 4, 6, and 12 h and filtered using a 0.45 μm membrane filter, with an equal volume of water replaced after each sampling. Again, the UDCA concentrations in the filtrates were analyzed using an LC-MS/MS system.

#### 2.4.3. Preparation of UDCA Formulation

A Yamato ADL311-A nozzle-type mini spray-dryer (Yamato Scientific Co., Ltd.; Tokyo, Japan) was employed to prepare pH-modified controlled-release UDCA formulations. Various ratios of UDCA (200 mg), HPMC (100, 200, or 600 mg), and NaOH (10 to 50 mg) or Na_2_CO_3_ (30 to 200 mg) were dissolved in 300 mL of 70% methanol. Each resulting solution was continuously stirred and transferred to a 0.4 mm pneumatic nozzle using a peristaltic pump and spray-dried to produce pH-modified controlled-release UDCA formulations. The spray-drying conditions are as follows: the inlet and outlet temperatures were set at 150 and 89–91 °C, respectively, with a feed rate of 2.0 mL/min. The drying air was maintained with a blow rate of 0.6 m^3^/min and a pressure of 0.15 Mpa for atomizing.

### 2.5. Characterization of UDCA Formulation

Morphological characteristics were observed using a field emission scanning electron microscope. The surface properties of the UDCA, Na_2_CO_3_, and HPMC samples, the physical mixture (PM), and the spray-dried formulation of UDCA:HPMC:Na_2_CO_3_ at the ratio of 200:600:150 (*w*/*w*/*w*) were analyzed using the Hitachi SU8000 cold field emission scanning electron microscope (FE-SEM) (Hitachi, Tokyo, Japan) at magnifications of 100–1000-fold. The analysis of the surface shape of the samples was conducted using a double-sided carbon tape attached to a platinum stub and sprayed with a powder sample. Then, a platinum coating was applied under a vacuum condition. Next, the sample was mounted onto a microscope. The sample had an operating pressure of 0.8 Pa, an acceleration voltage of 0.1–30 kV (0.1 kV per step), and an energy-dispersive X-ray spectroscopy detector system.

The X-ray diffraction (XRD) scanning of the UDCA, Na_2_CO_3_, and HPMC samples, the PM, and the spray-dried formulation of UDCA:HPMC:Na_2_CO_3_ at the ratio of 200:600:150 (*w*/*w*/*w*) was performed using an Empyrean X-ray diffractometer (Malvern Panalytical Ltd., Malvern, UK) using Cu Kα radiation at 40 mA and 40 kV. Data were obtained from 5–70° (2 thetas) with a step size of 0.02° and a scanning speed of 5°/min.

Differential scanning calorimetry (DSC) thermograms of the UDCA, Na_2_CO_3_, HPMC samples, the PM, and the spray-dried formulation of UDCA:HPMC:Na_2_CO_3_ at the ratio of 200:600:150 (*w*/*w*/*w*) were determined using a DSC Q2000 (TA Instruments, New Castle, DE, USA). Approximately 5 mg of a sample was placed in a closed aluminum pan and heated with a scanning rate of 5 °C/min from 10 to 250 °C, with nitrogen purging at 20 mL/min. The temperature scale was calibrated using indium.

The Fourier-transform infrared spectroscopy (FT-IR) spectra of the UDCA, Na_2_CO_3_, HPMC samples, the PM, and the spray-dried formulation of UDCA:HPMC:Na_2_CO_3_ at the ratio of 200:600:150 (*w*/*w*/*w*) were obtained in the spectral region of 4000–600 cm^−1^ with a resolution of 4 cm^−1^ and 64 scans using a Frontier FT-IR spectrometer (PerkinElmer, Norwalk, CT, USA) in the transmittance mode.

The solubility of UDCA was determined by first mixing the spray-dried powder of UDCA formulation, PM, and UDCA (all equivalent to 20 mg of UDCA) in 5 mL of distilled water and incubating the mixtures at 25 °C for 6 h. Then, the mixtures were filtered through a 0.45 μm membrane filter and diluted 100-fold in 50% methanol. UDCA concentration in the diluted filtrates was analyzed using an LC-MS/MS system.

Dissolution studies were conducted in 900 mL of distilled water for 12 h in a D-63150 dissolution test apparatus (Erweka, Heusenstamm, Germany) at 37 °C and 50 rpm using a paddle method (a type 2 USP dissolution method). Briefly, a spray-dried powder of UDCA formulation, PM, and UDCA (all equivalent to 20 mg of UDCA) were packaged into a hard gelatin capsule (size No. 0) and placed inside a sinker. A 1 mL aliquot of a medium was collected at 0, 1, 2, 4, 6, 12 h and filtered using a 0.45 μm membrane filter, with an equal volume of water replaced after each sampling. The UDCA concentrations in the filtrates were analyzed using an LC-MS/MS system.

For the final formulation (F7; UDCA:HPMC:Na_2_CO_3_ at the ratio of 200:600:150 (*w*/*w*/*w*)) and UDCA (all equivalent to 20 mg of UDCA), we performed the dissolution studies using fasted-state simulated gastric fluid (faSGF; 0.08 mM sodium taurocholate, 0.02 mM phospholipids, 34.2 mM sodium chloride; pH 1.2), fasted-state simulated intestinal fluid (faSIF; 3 mM sodium taurocholate, 0.75 mM phospholipids, 29 mM monobasic sodium phosphate, 106 mM sodium chloride; pH 6.5), and fed-state SIF (feSIF; 15 mM sodium taurocholate, 3.75 mM phospholipids, 144 mM sodium acetate, 203 mM sodium chloride; pH 6.0) [27] with the same protocol described above except for the use of bio-relevant medium.

### 2.6. Pharmacokinetic Study

All animal care and experimental procedures were approved by the Animal Care and Use Committee of Kyungpook National University (No. 2021-0029, 27 January 2021) and carried out in accordance with the National Institutes of Health Guidance for the care and the use of laboratory animals.

Male Sprague–Dawley rats (7–8 weeks old, weighing 225–270 g) were purchased from Samtaco Co. (Osan, Kyunggido, Korea) and housed in the facility to undergo 1 week of acclimatization. For oral administration of UDCA, rats were fasted for 16 h with free access to water before the pharmacokinetic study.

Four rats received UDCA (5 mg/kg/1 mL saline containing 10% DMSO) intravenously. For oral administration, four rats received UDCA (30 mg/kg/5 mL suspended in 0.5% methyl cellulose suspension) via oral gavage. Blood samples (approximately 100 μL) were collected via the jugular vein through the heparinized capillary tube (Heinz Herenz, Hamburg, Germany) at 0 (pre-dose), 0.25, 0.5, 1, 2, 4, 8, and 24 h post dose under isoflurane anesthesia (isoflurane vaporizer to 2% with oxygen flow at 0.8 L/min). Blood samples were centrifuged at 16,000× *g* for 1 min, and 30 μL aliquot of plasma was stored at −80 °C until required for UDCA analysis.

To investigate the effect of pH modulator on the UDCA pharmacokinetics, sixteen rats were randomly divided into 4 groups: group 1, 2, 3, and 4 (*n* = 4 for each group) received UDCA suspension in distilled water (pH 6.3), 50 mM citrate solution (pH 2.7), 30 mM NaOH solution (pH 11.4), and 100 mM NaOH solution (pH 12.4), respectively, at doses of 30 mg UDCA/kg/5 mL via oral gavage. Blood samples (approximately 100 μL) were collected via the jugular vein through the heparinized capillary tube at 0 (pre-dose), 0.25, 0.5, 1, 2, 4, 8, and 24 h post dose under isoflurane anesthesia. Blood samples were centrifuged at 16,000× *g* for 1 min, and 30 μL of plasma was stored at −80 °C until required for UDCA analysis.

Next, to compare the relative BA of UDCA formulations, twelve rats were randomly divided into 3 groups: group 1, 2, and 3 (*n* = 4 for each group) received UDCA, UDCA formulation F7, and F5 suspension, respectively, via oral gavage (all are equivalent to UDCA 20 mg/kg/5 mL in 0.5% methyl cellulose suspension). Blood samples (approximately 100 μL) were collected via the jugular vein through the heparinized capillary tube at 0 (pre-dose), 0.5, 1, 2, 4, 8, 24, 30, and 48 h post-dose under isoflurane anesthesia. Blood samples were centrifuged at 16,000× *g* for 1 min, and 30 μL of plasma was stored at −80 °C until required for UDCA analysis.

### 2.7. LC-MS/MS Analysis of UDCA

The LC-MS/MS method utilized an Agilent 6470 triple quadrupole LC-MS/MS system (Agilent, Wilmington, DE, USA) to analyze the concentrations of UDCA according to the previous methods with a slight modification [1,28,29,30,31]. The calibration standard solutions were prepared by evaporating the stock solution of UDCA and reconstituting with the same volume of activated charcoal treated rat blank plasma to make final concentrations of 10, 20, 50, 200, 500, 1000, 2000, 5000, and 10,000 ng/mL of UDCA. The quality control (QC) samples were made with the same protocols to make 30, 750, and 4000 ng/mL of UDCA. Then, 100 μL of naringenin solution was added to the 30 μL aliquot of calibration standards, QC samples, or rat plasma samples and mixed vigorously for 10 min, which was followed by centrifugation at 16,000× *g* for 10 min. The supernatants (100 μL each) were transferred to an autosampler vial, and 5 μL aliquots were injected into the LC-MS/MS system.

UDCA was separated on a Kinetex C18 column (75 × 4.6 mm, 2.6 μm particle size; Phenomenex, Torrance, CA, USA) with a mobile phase comprising of 0.1% formic acid in water: 0.1% formic acid in methanol = 25:75 (*v*/*v*) at a flow rate of 0.3 mL/min. The column and autosampler temperatures were 30 °C and 6 °C, respectively. The electrospray ionization (ESI) source settings were as follows: gas temperature 300 °C; gas flow 10 L/min; nebulizer pressure 35 psi; capillary voltage 4000 V; and nozzle voltage 500 V. Quantification was performed using multiple reaction monitoring in the negative ion mode with *m*/*z* 391.3 → 391.3 for UDCA (fragmentor 225 V, collision energy 45 V) and *m*/*z* 271.1 → 151.0 for IS (fragmentor 135 V, collision energy 15 V), respectively. The UDCA standard calibration curve was linear in the concentration range of 10–10,000 ng/mL, and the inter-day and intra-day precision and accuracy for UDCA was less than 15%.

### 2.8. Data Analysis

Pharmacokinetic parameters were calculated using WinNonlin (version 5.1; Pharsight, Certara, NJ, USA). with the non-compartmental analysis. The data are expressed as the means ± standard deviation for each group. Statistical analysis was performed using the Student’s *t*-test.

## 3. Results

### 3.1. pH-Dependent Solubility and Permeability of UDCA

First, we investigated the pH dependency in the solubility and permeability of UDCA. As shown in Figure 2A, UDCA solubility increased dramatically at pH 7.5 buffer compared with that at low pH range from 2.0 to 5.5. Considering that the pKa value of UDCA is 5.5 [32], the ionization of UDCA may contribute to the 6842-fold increase in its solubility at higher pH than its pKa value. However, UDCA exhibited similar permeability characteristics at pH 5.4 and 7.4 (Figure 2B). At pH 5.4, the absorptive, apical-to-basal permeability (P_app,AB_) of UDCA was calculated to be 2.5 × 10^−6^ cm/s and 4.1-fold higher than its basal-to-apical permeability (P_app,BA_). At pH 7.4, the P_app,AB_ of UDCA was 3.6 × 10^−6^ cm/s and 4.1-fold higher than P_app,BA_. In addition, the P_app,AB_ of UDCA at pH 7.4 was 1.5-fold higher than at pH 5.4. These results suggested that UDCA had moderate intestinal permeability and the limited involvement of efflux pumps in the absorption process [23,24]; therefore, UDCA could be readily absorbed at pH 5.4 and more efficiently at pH 7.4.

Alteration in the TEER values before and after the UDCA permeability study were 6.0 ± 0.9% at pH 5.4 (before: 418 ± 10 Ω·cm^2^ and after: 392 ± 20 Ω·cm^2^, respectively) and 7.5 ± 1.5% at pH 7.4 (before: 417 ± 9.9 Ω·cm^2^ and after: 385 ± 12 Ω·cm^2^, respectively). The results suggested that the presence of 20 µM UDCA did not alter the cell integrity. Moreover, P_app,AB_ and P_app,BA_ values of propranolol, the high-permeability marker, were 25.1 ± 2.5 × 10^−6^ cm/s and 20.2 ± 0.8 × 10^−6^ cm/s, respectively. P_app,AB_ and P_app,BA_ values of atenolol, the low-permeable and paracelluar marker, were 0.53 ± 0.02 × 10^−6^ cm/s and 0.56 ± 0.05 × 10^−6^ cm/s, respectively. The results were similar to the previous reports and suggested the feasibility of our Caco-2 permeability study [24].

### 3.2. Oral Bioavailability (BA) of UDCA

Despite the prevalent use of UDCA, its oral BA remains poorly understood. Therefore, we investigated the oral BA of UDCA in rats (Figure 3A). From the plasma profile of UDCA following intravenous and oral administration, the absolute BA of UDCA was calculated to be 15.2%. Based on the pH dependency in the solubility and permeability of UDCA (Figure 2A), we next investigated the effect of pH modulators, such as sodium citrate or sodium hydroxide, on the pharmacokinetics of UDCA. The coadministration of 50 mM sodium citrate with UDCA in an oral suspension at pH 2.7 showed similar AUC values compared with the oral administration with UDCA alone (Table 1). However, the decreased pH in the oral suspension using sodium citrate showed a double-peak phenomenon at about 8 h (Figure 3B), which was likely because the solubility and absorption of UDCA in the upper intestinal tract, i.e., the stomach and duodenum, under low pH conditions were reduced and its solubility and absorption at the lower part of the intestinal tract with elevated pH were recovered.

However, adding 30 mM sodium hydroxide to achieve a pH of 11.4 in the oral suspension did not alter the pharmacokinetics of UDCA (Figure 3B and Table 1). Even when we increased the sodium hydroxide concentration to 100 mM to achieve a pH of 12.4 in the oral suspension, the pharmacokinetics of UDCA was not altered (Figure 3B and Table 1). The results suggested that pH modulation for a short period might not improve the oral BA of UDCA. Therefore, the sustained release of an alkalizer appears necessary to enhance the solubility of UDCA for a long period and to increase its oral BA.

### 3.3. Optimization of pH-Modified Controlled Release Formulation of UDCA

The content of HPMC, which was used for the controlled-release modulator, was determined by preparing spray-dried powders consisting of UDCA and NaOH with varying proportions of HPMC from 100 to 600 mg (Step I stage in Table 2; F1–F3). In the case of F3, the dissolution rate of UDCA increased gradually for 4 h, whereas the dissolution rate of UDCA from F1 and F2 showed a dramatic increase within 1 h. The dissolution of UDCA from UDCA crystalline powder was much lower compared with that from the formulation (Figure 4A). Therefore, we selected the ratio of UDCA and HPMC at 200 mg: 600 mg. However, the amount of dissolved UDCA was less than 22%, suggesting the incomplete dissolution of UDCA. Therefore, we increased the alkalizing agent and dissolution time for further optimization.

The optimal alkalizing agent was determined by measuring UDCA solubility with increasing amounts of NaOH and Na_2_CO_3_ (Figure 4B). The effect of NaOH on the UDCA solubility was about 3-fold greater than the equal amount of Na_2_CO_3_. Hence, we compared the UDCA solubilities of the spray-dried powder formulations of UDCA with NaOH or Na_2_CO_3_ (Step II stage in Table 2). All spray-dried formulations (F4–F7) demonstrated similar solubility in the range of 8–9 mg/mL, which was higher than that from UDCA itself; in addition, using Na_2_CO_3_ at larger than three times the amount of NaOH appeared to achieve sufficient solubility (Figure 4C). Despite the similar solubilities among the formulations from F4 to F7, the dissolution profiles of these formulations differed depending on their compositions. In the case of F4 and F5, the dissolution rate of UDCA increased up to 2 h, but the dissolution rate gradually increased, reaching a steady state during the 2–12 h period (Figure 4D). This profile was consistent with previous results (F2 in Figure 4A), but the dissolution amount in F4 and F5 was much greater than F2 and UDCA itself, which was due to the use of a greater amount of alkalizing agent. In the case of F6 and F7, the dissolution rate of UDCA increased up to 6 h, reaching a steady state during the 6–12 h period. The amount of dissolved UDCA was much greater in the case F6 and F7 (with 600 mg HPMC) and in the case of NaOH (F4 and F6), suggesting the favorable role of controlled-release formulations and that the amount of the alkalizing agent should be optimized further.

Next, we measured the solubility and dissolution depending on the increasing amount of alkalizing agent Na_2_CO_3_ (Step III stage in Table 2). As shown in Figure 5, the solubility of UDCA was increased with the increasing amount of Na_2_CO_3_ up to 150 mg (F8, F9, F7) but decreased at 200 mg of Na_2_CO_3_ (F10) (Figure 5A). Similarly, the dissolution rate of UDCA increased with the increasing amount of Na_2_CO_3_ up to 150 mg (F8, F9, F7) but slightly decreased when 200 mg of Na_2_CO_3_ was used (F10) (Figure 5B). Therefore, we determined the optimal formulation of UDCA as that with the UDCA:HPMC:Na_2_CO_3_ ratio of 200:600:150 (*w*/*w*/*w*).

Because of the hygroscopic or deliquescent nature of the alkalizing agent, we measured the alterations in the weight and UDCA content in the spray-dried powder after 3 weeks of storage at 30 °C and 65% relative humidity (RH) condition. F6 exhibited morphological changes due to the deliquescent effect of NaOH (Figure 6A). The weight change was calculated to be about 5–7% in all formulations (Figure 6B). However, the UDCA content in F4 and F6, which used NaOH as the alkalizing agent, decreased about 30% (Figure 6C). In the case of F5 and F7, which used Na_2_CO_3_, their surface morphologies or UDCA contents remained unchanged (Figure 6). These results suggested that Na_2_CO_3_ would be a better alkalizing agent than NaOH for stabilizing a formulation.

### 3.4. Characterization of pH-Modified Controlled Release Formulation of UDCA

UDCA and Na_2_CO_3_ appeared as irregular particles with a broad range of sizes at 0.25–3.32 μm for UDCA and less than 200 μm for Na_2_CO_3_. Meanwhile, HPMC appeared as large blocks with a particle size of 40–100 μm (Figure 7A). In addition, the PM at the UDCA:HPMC:Na_2_CO_3_ ratio of 200:600:150 (*w*/*w*/*w*) exhibited similar morphologies to those of HPMC and Na_2_CO_3_ (Figure 7A, PM). However, UDCA formulations F5 and F7 appeared similarly round-shaped with a particle size of around 20 μm or less regardless of the composition of the formulations (Figure 7A; F5 and F7). 

In addition, the XRD pattern analysis (Figure 7B) displayed the eventual structural change in UDCA and Na_2_CO_3_ by the spray-drying process. The multiple diffraction angles around 5 to 50 thetas in UDCA (i.e., 9.4°, 11.8°, 12.3°, 13.4°, 13.8°, multiple peaks around 15.2°~16.6°, 17.5°, 19.6°, 20.8°, 21.3°, 22.4°, and 24.8°) were pretty consistent with the previous reports [33,34], and these peaks disappeared in F5 and F7. Similarly, the multiple diffraction angles around 10 to 50 thetas in Na_2_CO_3_ appeared at 16.5°, 19.8°, 22.1°, 25.3°, 26.3°, 30.7°, 32.4°, 35.0°, 38.0°, and 47.5°, and these peaks disappeared in F5 and F7. However, the peaks from UDCA and Na_2_CO_3_ remained in UDCA-PM. The two major diffraction peaks of HPMC at 8 and 20 thetas remained in the UDCA formulation as well as PM (Figure 3B). Collectively, the structural nature of the spray-dried formulations F5 and F7 appeared to change into amorphous states.

Next, the DSC thermal behavior of F5 and F7 was compared to UDCA alone and UDCA-PM. The DSC thermogram indicated that UDCA had a sharp peak at 204.8 °C, and Na_2_CO_3_ had three peaks at 34.22, 62.05, and 101.79 °C. Consistent with the XRD patterns, HPMC did not exhibit obvious glass transition peaks in the range of 10–250 °C, which was likely due to its amorphous structure (Figure 7C). The DSC thermogram of PM displayed two peaks likely belonging to UDCA (at 204.9 °C) and Na_2_CO_3_ (88.5–120 °C). Moreover, the DSC patterns of the spray-dried formulations F5 and F7 exhibited amorphous characteristics, showing no obvious glass transitions (Figure 7C).

Then, the FT-IR patterns of UDCA, Na_2_CO_3_, HPMC, PM, F5, and F7 were compared (Figure 7D). PM exhibited characteristic peaks of UDCA, Na_2_CO_3_, and HPMC, suggesting their mixed physical states. However, new bands were observed at 1645 cm^−1^ and 1400 cm^−1^ (blue triangles in Figure 7D) for F5 and F7 instead of a sharp peak at 1714 cm^−1^ (black arrow in Figure 7D) shown in UDCA and PM. These peak changes, also detected in previous formulations of UDCA-loaded nano vehicles coated with Eudragit S100 [35], are known to be caused by the asymmetric and symmetric vibrations of the COO^−^ group in UDCA [35,36]. In addition, all the characteristic bands from UDCA, Na_2_CO_3_, and HPMC were found and overlapped in the spectrum of F5 and F7, which indicated that UDCA molecules are stabilized and co-existed with Na_2_CO_3_ in the sphere of HPMC through electrostatic interaction without any chemical structural changes. Taken together, these data in Figure 7 suggest that UDCA formulations F5 and F7 stayed in the amorphous state without molecular interaction among UDCA, Na_2_CO_3_, and HPMC after the spray-drying process.

Afterward, we compared the solubility and dissolution of F7 with that of UDCA itself and the PM of the same composition as F7 (Figure 8A,B). The solubility of F7 increased by 3200-fold compared with UDCA and 1.5-fold with PM. The results suggested formulation with an alkalizer greatly increased UDCA solubility, and the amorphous state of F7 even contributed to its solubility. The differences in solubility were consistent with the dissolution profile of F7, PM, and UDCA. We also measured the stability of F7 by measuring the solubility and dissolution profile of UDCA in the F7 stored in a thermo-hygrostat at 30 °C and 65% RH for 3 months. The solubility and dissolution profile of F7 was not changed by the storage (Figure 8C,D), indicating the stability of F7.

Next, we compared the dissolution profile of F7 with that of UDCA itself using bio-relevant medium (i.e., FaSGF (pH 1.2), FaSIF (pH 6.8), and FeSIF (pH 6.5)) (Figure 9). The dissolution of UDCA was very low for 4 h in FaSGF (pH 1.2) from both UDCA and F7 groups. The low pH condition may contribute to the limited dissolution. However, in both FaSIF (pH 6.8) and FeSIF (pH 6.5), the dissolution rate of UDCA from F7 was much greater than that from UDCA only. The UDCA dissolution rate from the UDCA group plateaued at 1 h, while the dissolution rate from F7 reached plateau around 6 h. The results confirmed the pH-modified extended release and higher dissolution of UDCA from F7 compared with the UDCA group in both FaSIF (pH 6.8) and FeSIF (pH 6.5) (Figure 9B,C) as well as in distilled water (Figure 8B).

### 3.5. Pharmacokinetics of UDCA from UDCA Formulation

To investigate the effect of the pH-modified extended and increased release of UDCA in vitro dissolution test on the oral absorption of UDCA, we next compared the pharmacokinetics of UDCA and spray-dried formulations of F5 and F7 (equivalent to 20 mg/kg UDCA) after oral administration (Figure 10). After the oral administration of F7_,_ the plasma concentration–time profile of UDCA was higher than those of UDCA and F5. Consequently, the AUC was significantly greater in the F7 group than the control UDCA and F5 groups without significant alterations in C_max_, t_1/2_, and MRT values (Table 3). The relative bioavailability of UDCA in F7 was 251% compared to the UDCA group. These data suggest an increased dissolution and solubility of UDCA following the administration of F7 compared with UDCA alone.

## 4. Discussion

In this study, we have confirmed that the solubility and permeability of UDCA are increased under higher pH conditions (more than 6.0). However, the concomitant administration of an alkalizing agent (NaOH 30 mM and 100 mM) did not increase UDCA’s oral bioavailability. The results suggested that UDCA absorption could not be improved because the pH was temporarily increased during UDCA administration. Thus, the prolonged dissolution of UDCA and alkalizer, that could maintain a high solubility of UDCA under higher pH conditions at its absorption site, could contribute to the enhanced oral BA of UDCA.

Lipid-based carriers with low BA therapeutic drugs could augment intestinal absorption by enhancing the M-cell-mediated uptake, transcellular, and paracellular pathway and by decreasing its efflux mechanism [25,37,38]. UDCA showed moderate absorptive permeability (2.5 × 10^−6^ cm/s at pH 5.4 and 3.6 × 10^−6^ cm/s at pH 7.4) and 4.1-fold higher absorptive permeability than secretory permeability (Figure 2). It is suggested that UDCA could be readily absorbed at pH 5.4 and more efficiently at pH 7.4 without involving efflux pumps. In addition, the absorption of UDCA was known to occur by passive diffusion as well as active bile acid transporter [13,39]. Therefore, the formulation strategy was focused on maintaining a higher pH at the dissolution site by creating an extended-release UDCA formulation with an alkalizing agent.

To achieve the extended release of UDCA, we applied HPMC in this study, since HPMC has been widely used as the conventional sustained release matrix system [21,22]. The formation of an HPMC gel layer upon contacting the UDCA-loaded formulation to the medium during the dissolution is considered important because it may impact the overall drug release [21,40]. Hence, the dissolution and solubility of UDCA formulations with various ratios of an extended releasing excipient, HPMC, and alkalizing agents, Na_2_CO_3_ or NaOH, were compared to optimize the composition of the UDCA formulation. Subsequently, Na_2_CO_3_ was selected based on the least moisture absorption and stability of UDCA in the spray-dried UDCA formulation with the UDCA:HPMC:Na_2_CO_3_ ratio of 200:600:150 (*w*/*w*/*w*) during storage. This formulation, with spherical morphology and a particle size of around 20 μm, remained in amorphous states after the spray-drying process. In addition, the optimized formulation showed an increased solubility and elevated dissolution profile compared with UDCA itself as well as PM. Moreover, the formulation remained stable for more than 3 months in storage at 30 °C and 65% RH. The similarity factors calculated from the dissolution profile of the stored UDCA formulation were 63.9%, 65.7%, and 52.9% at 1, 2, and 3 months, respectively, compared with the control formulation. The dissolution profile of UDCA from the storage of 1, 2, and 3 months was considered to be similar because the similarity factors were greater than 50 [41,42]. Finally, to confirm the sustained release of spray-dried UDCA formulation with the UDCA:HPMC:Na_2_CO_3_ ratio of 200:600:150 (*w*/*w*/*w*) in the biorelevant media, we performed the dissolution test of UDCA formulation in FaSGF (pH 1.2) for 4 h and in FaSIF (pH 6.8) and FeSIF (pH 6.5) for 12 h (Figure 9). The dissolution of UDCA was less than 10% for 4 h in FaSGF but increased steadily for 6 h, and about 80–95% of UDCA loaded in the formulation was dissolved for 12 h. The results confirmed the pH-modified extended release and higher dissolution of UDCA from the spray-dried UDCA formulation with the UDCA:HPMC:Na_2_CO_3_ ratio of 200:600:150 (*w*/*w*/*w*) compared with UDCA itself (about 30%) in FaSIF (pH 6.8) and FeSIF (pH 6.5). In addition, the in vitro dissolution profile correlated with the in vivo absorption, since this final formulation showed enhanced oral BA by 251% compared with UDCA itself.

Regarding the number of animals and statistical power, the pharmacokinetic parameters such as C_max_ and AUC values of UDCA from the F7 group were significantly greater than the UDCA control group using four rats per group (Table 3). The statistical power of the existing AUC results were calculated as 98% with a significance level of 0.05 in the post hoc power analysis using our pharmacokinetic data from the F7 and UDCA groups [43]. In addition, the minimum sample size was estimated to three rats per group using clinical calculators (clincalc.com; accessed on 7 April 2022) for adequate study power (significance level of 0.05 and statistical power of 80%) of our UDCA pharmacokinetic data [44]. The estimation indicated that our study design was adequate to detect statistical significance of AUC values of UDCA between two groups.

## 5. Conclusions

The present study confirmed the importance of higher pH for the absorption of UDCA. That is, the aqueous solubility of UDCA greatly increased by 6842-fold at pH 7.5 compared with that at pH 5.5. The absorptive permeability of UDCA was 4.1-fold higher than the secretory permeability, and it increased 1.5-fold at pH 7.4, suggesting the favorable and enhanced absorption of UDCA at higher pH condition. For this, we have successfully formulated a solid dispersion powder formulation of UDCA using Na_2_CO_3_ as the alkalizing agent and HPMC as the release-modifying agent. The optimized pH-modified controlled-release UDCA formulation was prepared at the UDCA:HPMC:Na_2_CO_3_ ratio of 200:600:150 (*w*/*w*/*w*) using a spray-drying method. The formulation showed a sustained dissolution profile for 12 h and an increased UDCA solubility due to the continuous release of Na_2_CO_3_. In addition, this pH-modified extended-release UDCA formulation maintained a higher UDCA concentration than UDCA by itself or the non-extended-release formulation of UDCA for up to 48 h, consequently increasing the AUC significantly and effectively enhancing the oral bioavailability by 251% in rats. This study emphasized the importance of pH-modified extended-release formulation for the sustained release of UDCA and alkalizing agent together to maintain the higher pH condition at the UDCA releasing site for the better absorption of UDCA.

## Figures and Tables

**Figure 1 pharmaceutics-14-01037-f001:**
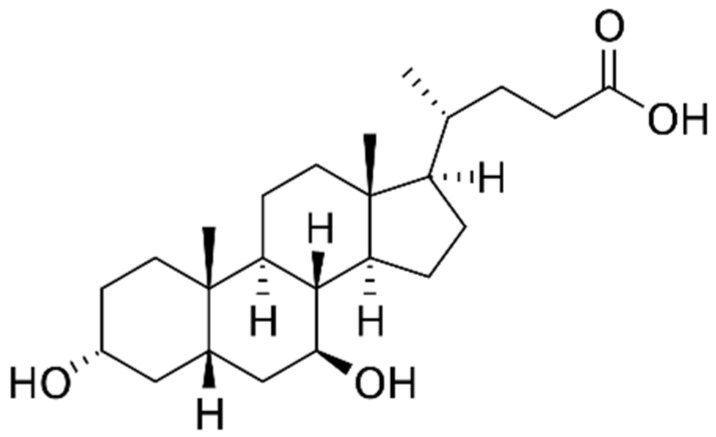
Chemical structure of ursodeoxycholate (UDCA).

**Figure 2 pharmaceutics-14-01037-f002:**
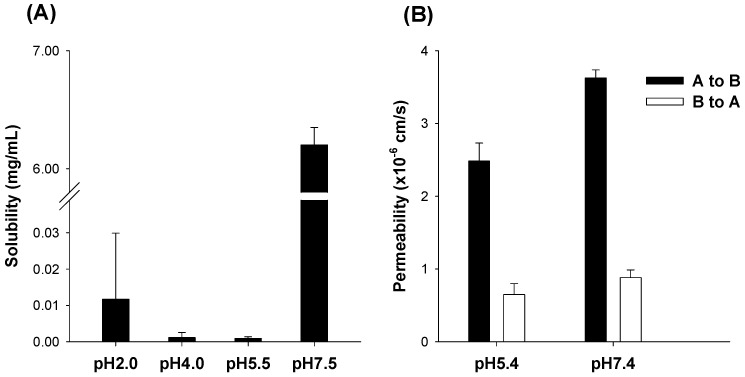
(**A**) The solubility of UDCA was measured under various pH conditions. (**B**) The apparent permeability (P_app_) of 20 µM UDCA was measured at pH 5.4 and 7.4 in Caco-2 cells. Each bar represents the mean ± standard deviation (*n* = 3).

**Figure 3 pharmaceutics-14-01037-f003:**
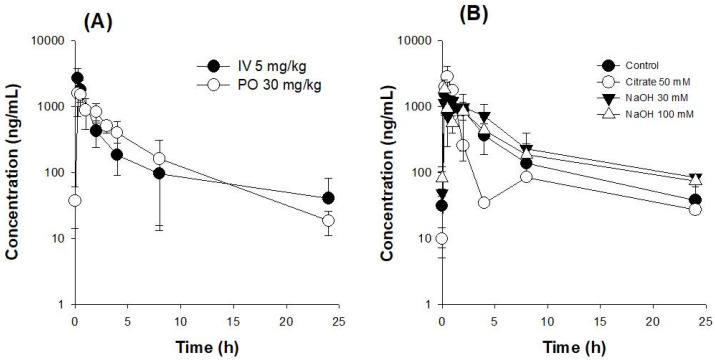
(**A**) Plasma concentration–time profile of UDCA following the intravenous injection of 5 mg/kg UDCA or oral administration of 30 mg/kg UDCA in rats. (**B**) The effect of pH modulators, such as sodium citrate or sodium hydroxide, on the plasma concentration–time profile of UDCA following oral administration of 30 mg/kg of UDCA in rats. Each data point represents the mean ± standard deviation (*n* = 4).

**Figure 4 pharmaceutics-14-01037-f004:**
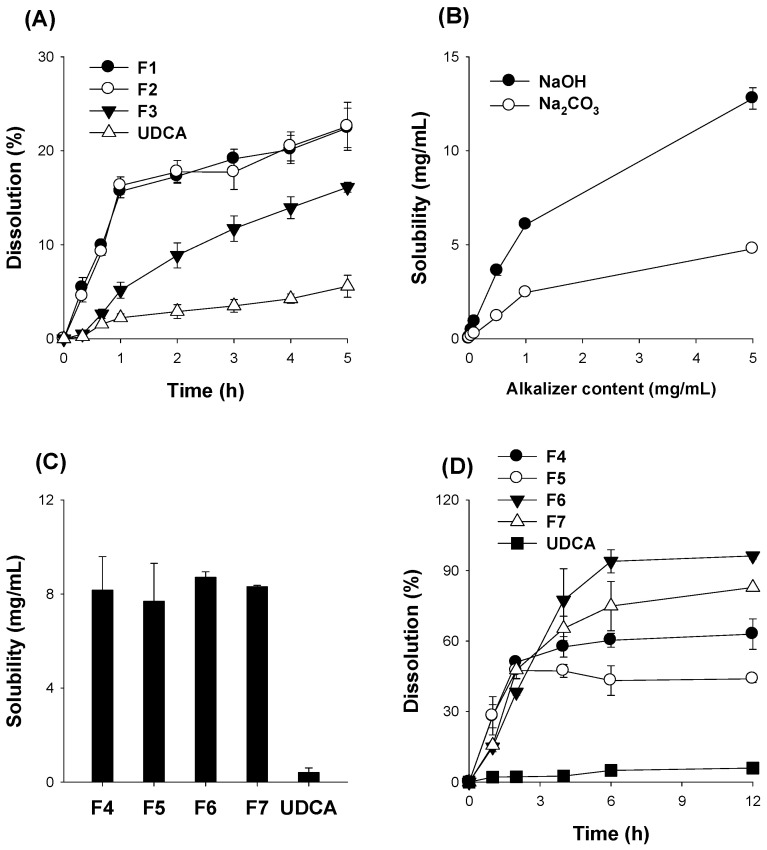
(**A**) Effect of HPMC content on the dissolution profile of UDCA in the spray-dried powder formulations with varying HPMC content (F1–F3) and UDCA. (**B**) Effect of increasing amounts of NaOH and Na_2_CO_3_ on UDCA solubility. (**C**) The solubility and (**D**) the dissolution profile of UDCA in the spray-dried powder formulations with varying ratios of UDCA:HPMC:alkalizing agent (F4–F7, Table 2) and UDCA itself. Each data point represents the mean ± standard deviation (*n* = 3).

**Figure 5 pharmaceutics-14-01037-f005:**
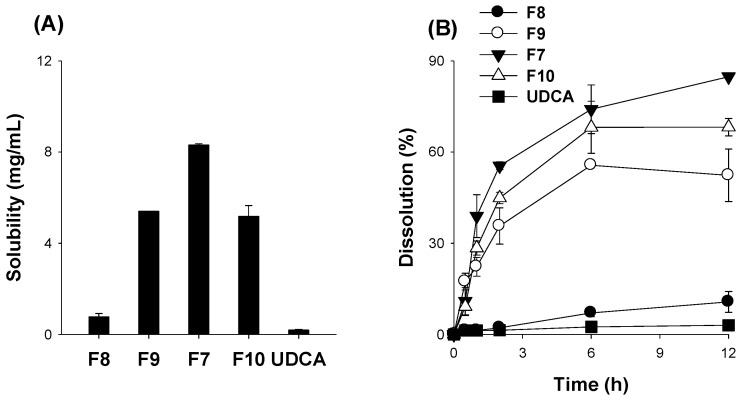
Effect of Na_2_CO_3_ content on (**A**) the solubility and (**B**) the dissolution profile of UDCA in spray-dried powder formulations with varying amounts of the alkalizing agent Na_2_CO_3_ (F7–F10, Table 2) and UDCA itself. Each data point represents the mean ± standard deviation (*n* = 3).

**Figure 6 pharmaceutics-14-01037-f006:**
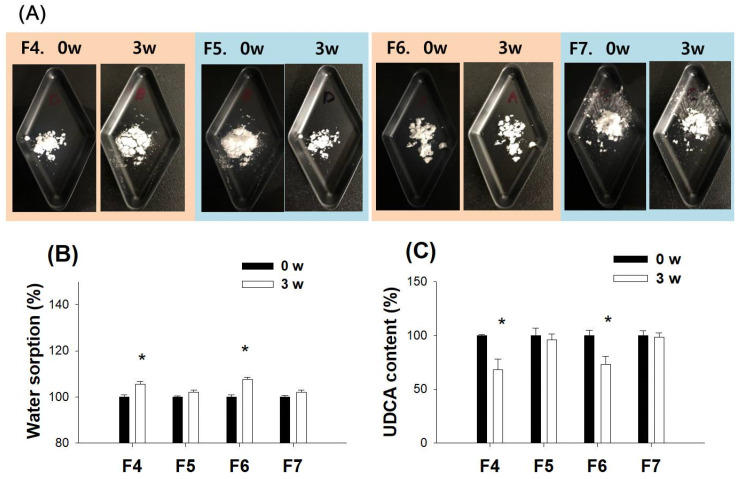
(**A**) Representative photos for the effect of storage at 30 °C and 65% RH for 3 weeks on the UDCA formulations (F4–F7). The effect of the storage on (**B**) the weight and (**C**) UDCA content (%) of the UDCA formulations (F4–F7). Each data point represents the mean ± standard deviation (*n* = 3). * *p* < 0.05, compared with a group with 0 weeks of storage using the Student’s *t*-test.

**Figure 7 pharmaceutics-14-01037-f007:**
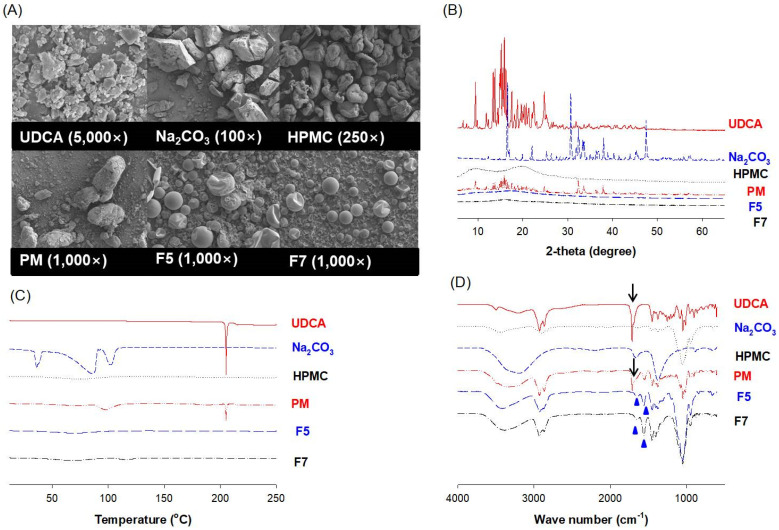
(**A**) Scanning electron microscope (SEM) images of UDCA (5000×), Na_2_CO_3_ (100×), HPMC (500×), physical mixture (PM, 1000×) with the UDCA:Na_2_CO_3_:HPMC ratio of 200:600:150 (*w*/*w*/*w*), and the spray-dried formulations F5 (1000×) and F7 (1000×). (**B**) The X-ray diffraction (XRD) patterns, (**C**) differential scanning calorimetry (DSC) thermograms, and (**D**) Fourier-transform infrared spectroscopy (FT-IR) spectrometry of UDCA, Na_2_CO_3_, HPMC, PM, F5, and F7. The compositions of PM, F5, and F7 are listed in Table 2.

**Figure 8 pharmaceutics-14-01037-f008:**
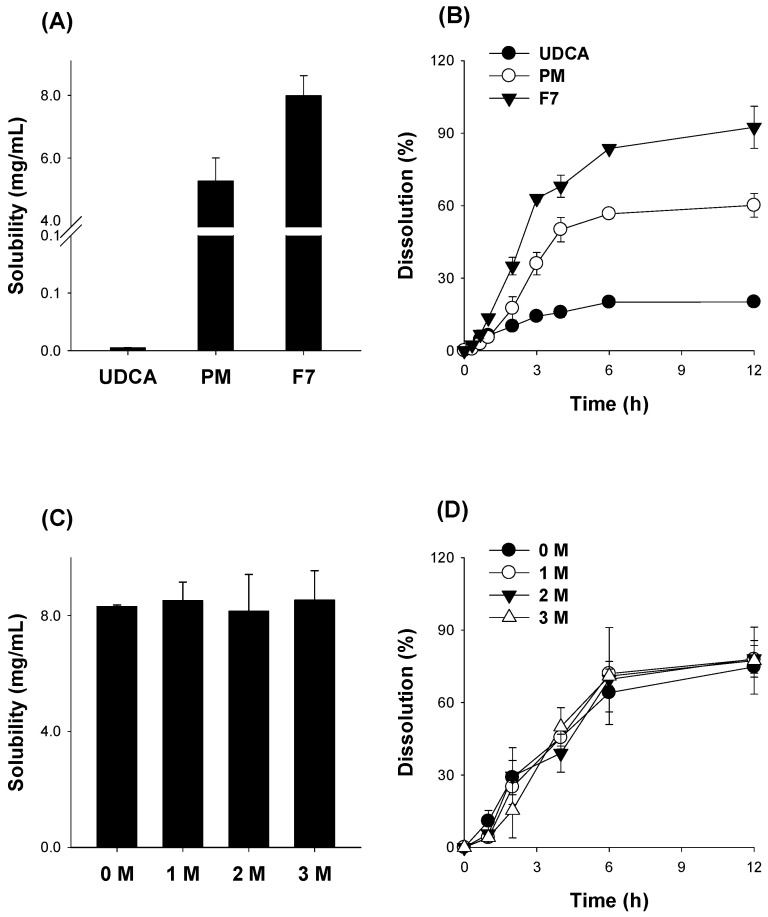
(**A**) The solubility and (**B**) the dissolution profile of UDCA, physical mixture (PM) with the ratio of UDCA:Na_2_CO_3_:HPMC = 200:600:150 (*w*/*w*/*w*), and spray-dried formulation F7. Effect of storage at 30 °C and 65% RH on (**C**) the solubility and (**D**) the dissolution profile of spray-dried formulation F7. Each data point represents the mean ± standard deviation (*n* = 6).

**Figure 9 pharmaceutics-14-01037-f009:**
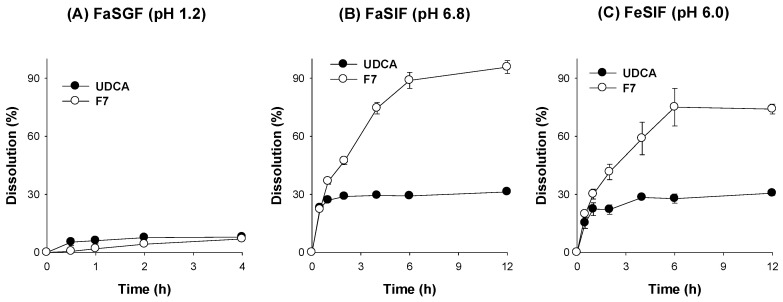
The dissolution profile of UDCA and spray-dried formulation F7 (UDCA:Na_2_CO_3_:HPMC = 200:600:150 (*w*/*w*/*w*) in (**A**) fasted-state simulated gastric fluid (faSGF; pH 1.2), (**B**) fasted-state simulated intestinal fluid (faSIF; pH 6.8), and (**C**) fed-state simulated intestinal fluid (feSIF; pH 6.0). Each data point represents the mean ± standard deviation (*n* = 4).

**Figure 10 pharmaceutics-14-01037-f010:**
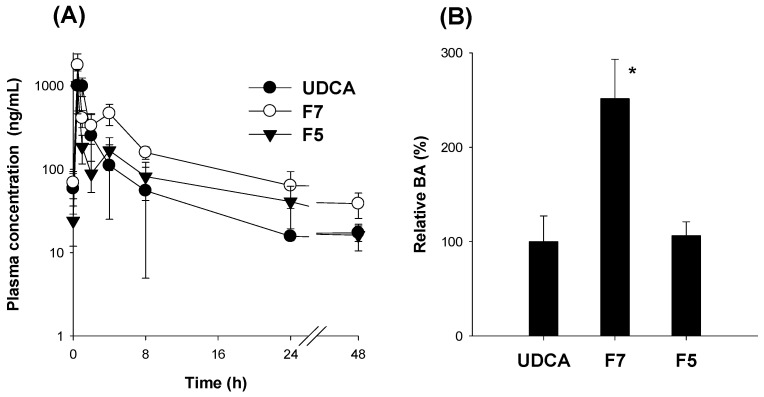
(**A**) Plasma concentrations and (**B**) relative BA of UDCA after the single oral administration of UDCA alone and UDCA formulations F5 and F7 (all equivalent to 20 mg/kg UDCA) in rats. The data point represents the mean ± standard deviation (*n* = 4). * *p* < 0.05, compared with UDCA control group using the Student’s *t*-test.

**Table 1 pharmaceutics-14-01037-t001:** Pharmacokinetic parameters of UDCA in rats.

	Treatment	IV (UDCA 5 mg/kg)		PO (UDCA 30 mg/kg)	
Parameters					
C_o_ (μg/mL)		4.17 ± 2.31			
C_max_ (μg/mL)				1.76 ± 0.84	
T_max_ (h)				0.44 ± 0.13	
AUC_24h_ (μg∙h/mL)		5.91 ± 3.19		5.42 ± 1.84	
AUC_∞_ (μg∙h/mL)		7.35 ± 5.29		5.72 ± 2.10	
t_1/2_ (h)		7.31 ± 2.11		5.45 ± 2.96	
MRT (h)		4.86 ± 2.80		5.68 ± 2.19	
Absolute BA (%)				15.2	
	Treatment	PO (UDCA 30 mg/kg)			
Parameters		Control (pH 6.3)	+Na citrate 50 mM (pH 2.7)	+NaOH 30 mM (pH 11.4)	+NaOH 100 mM (pH 12.4)
C_max_ (μg/mL)		1.81 ± 0.73	2.86 ± 1.11	1.33 ± 0.13	1.90 ± 0.69
T_max_ (h)		0.45 ± 0.11	0.42 ± 0.14	0.75 ± 0.43	0.69 ± 0.88
AUC_24h_ (μg∙h/mL)		5.77 ± 1.67	4.39 ± 1.65	6.78 ± 3.05	6.07 ± 2.50
AUC_∞_ (μg∙h/mL)		6.06 ± 2.01	4.63 ± 2.05	7.63 ± 2.84	6.66 ± 2.89
t_1/2_ (h)		4.26 ± 1.18	4.23 ± 2.04	4.61 ± 2.31	5.56 ± 2.99
MRT (h)		5.26 ± 2.50	3.99 ± 3.63	6.21 ± 2.91	6.89 ± 3.69
Relative BA (%)		100	76.4 ± 33.8	125 ± 46.8	109 ± 47.6

IV; intravenous injection, PO; per oral administration, C_o_; initial plasma concentration following intravenous injection of UDCA; C_max_, maximum plasma concentration following oral administration of UDCA; T_max_: time to reach C_max_; AUC_24h_ or AUC_∞_, area under the curve from zero to 24 h or from zero to infinity, respectively; t_1/2_, half-life; MRT, mean residence time; absolute BA, absolute BA (dose normalized AUC_IV_/dose normalized AUC_PO_ × 100); relative BA, relative bioavailability (AUC_treatment_/AUC_control_ × 100). Data represent the means ± standard deviation (*n* = 4).

**Table 2 pharmaceutics-14-01037-t002:** The content of spray-dried powder formulation.

Step	Formulations	UDCA (mg)	HPMC (mg)	NaOH (mg)	Na_2_CO_3_ (mg)
Step I	F1	200	100	10	
	F2	200	200	10	
	F3	200	600	10	
Step II	F4	200	200	50	
	F5	200	200		150
	F6	200	600	50	
	F7	200	600		150
Step III	F8	200	600		30
	F9	200	600		90
	F7	200	600		150
	F10	200	600		200
	PM	200	600		150

**Table 3 pharmaceutics-14-01037-t003:** Pharmacokinetic parameters of UDCA after the single oral administration of UDCA alone and UDCA formulations F5 and F7 (all equivalent to 20 mg/kg UDCA) in rats.

Parameters	UDCA	F7	F5
C_max_ (μg/mL)	1.19 ± 0.39	1.79 ± 0.65	1.02 ± 0.56
T_max_ (h)	0.87 ± 0.25	0.5 ± 0.0 *	0.5 ± 0.0 *
AUC_48h_ (μg∙h/mL)	3.03 ± 0.99	6.53 ± 0.95 *	3.03 ± 0.75
AUC_∞_ (μg∙h/mL)	3.34 ± 0.90	8.39 ± 1.53 *	3.54 ± 0.49
t_1/2_ (h)	13.7 ± 8.93	14.2 ± 3.55	15.6 ± 5.12
MRT (h)	14.9 ± 6.3	18.2 ± 5.12	20.3 ± 6.97
Relative BA (%)	100 ± 27	251 ± 45.9	106 ± 14.7

C_max_, maximum plasma concentration; T_max_, time to reach C_max_; AUC_48h_ or AUC, area under the curve from zero to 484 h or from zero to infinity, respectively; t_1/2_, half-life; MRT, mean residence time; relative BA, relative bioavailability (AUC_F7 or F5/_AUC_UDCA_ × 100). *: *p* < 0.05, statistically significant compared with UDCA control group by Student’s *t*-test. Data represent the means ± standard deviation (*n* = 4).

## Data Availability

The data presented in this study are available upon request.

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
