# Peer review of "Enhancing Dissolution and Oral Bioavailability of Ursodeoxycholic Acid with a Spray-Dried pH-Modified Extended Release Formulation"

_pharmaceutics, 2022, doi:10.3390/pharmaceutics14051037_

Round 1
Reviewer 1 Report
Please see the attachment.

Author Response
Thank you for the reviewer’s valuable comments.
Our point by point responses to the reviewer’s comments were attached.

Reviewer 2 Report
The authors describe a thorough study of the formulation, in vitro characterization and pharmacokinetic evaluation a bile acid oral formulation. Although the study is interesting and deserves publication, several issues should be resolved first:
- Abstract:
- suggest to remove statement "Thus, maintaining higher pH at the absorption site is important for increasing its oral bioavailability.", as it does not follow from the previous sentence.
2. Materials and methods
- Permeability measurements - caco2-cells can be adversely affected by bile salts, due to the lack of protective mucous layer. Was the viability and integrity of the Caco2-cell layer assessed before and after the permeation experiments (e.g. TEER values, permeation of a non-permeating markers, etc.)?
- Spray-drying - what was the atomizing gas used? What was its flow rate? What was the flow rate of the aspirating gas (100% is not clear enough)?
- Dissolution - distilled water was used for the dissolution studies, instead of biorelevant media with closer composition to the real in vivo situation, where the bile salts and phospholipids present in the intestinal fluids will greatly affect the obtained results for UDCA. These experiments need to be performed and the validity of the current conclusions should be checked
3. Results
- Dissolution data (e.g. Fig. 4 and 5): the dissolution profile of the reference crystalline UCDA, and, possible a reference marketed product should be displayed, in order to evaluate the performance of the new formulations.
Author Response

(The authors gave the same response as above.)

Round 2
Reviewer 2 Report
The authors have addressed my comments and the manuscript is suitable for publication.